# Burnout and Time Perspective of Blue-Collar Workers at the Shipyard

**DOI:** 10.3390/ijerph17186905

**Published:** 2020-09-21

**Authors:** Sarah Detaille, Adela Reig-Botella, Miguel Clemente, Jaime López-Golpe, Annet De Lange

**Affiliations:** 1Department of Human Resource Management, HAN University of Applied Sciences in Nijmegen, 6525EJ Nijmegen, The Netherlands; sarah.detaille@han.nl (S.D.); annet.delange@han.nl (A.D.L.); 2Department of Psychology, Universidade da Coruna, 15701 A Coruña, Spain; miguel.clemente@udc.es (M.C.); jaime.lopez.golpe@gmail.com (J.L.-G.); 3Faculty of Psychology, Open University Heerlen, 6419 AT Heerlen, The Netherlands; 4Norwegian School of Hotel Management, University of Stavanger, 4021 Stavanger, Norway; 5Faculty of Psychology, Norwegian University of Science and Technology, 7491 Trondheim, Norway

**Keywords:** ageing, blue-collar workers, cross-sectional, occupational health, risk burnout, shipyard, time perspective

## Abstract

*Background:* The aim of the research was to investigate the association between time perspective in relation to burnout and successful ageing of blue-collar workers with physically highly demanding work and low autonomy. Shipyard blue-collar workers usually do predominantly manual labor versus white-collar workers, whose jobs do not usually involve physical work. *Methods:* 497 participants workers in a shipyard in the north of Spain. Ages were between 20 and 69 (M = 46.62, SD = 10.79). We used the Zimbardo Time Perspective Instrument (ZTPI), Spanish version, and the Maslach Burnout Inventory–General Survey (MBI-GS). *Results:* Emotional exhaustion factor obtained a coefficient of 0.97; cynicism factor of 0.83; and professional efficacy factor of *p* = 0.86. The mean of the three reliability coefficients was 0.887. With respect to the five factors of the ZTPI questionnaire: the negative past factor obtained a coefficient of *p* = 0.91; that of positive past *p* = 0.81; the present hedonistic of 0.878; the future of *p* = 0.83; and the fatalistic present of *p* = 0.90. The mean of the five coefficients, was *p* = 0.86. *Conclusions:* Within shipyard workers, burnout is associated with a negative past and negative future time perspective. This makes shipyard workers at a higher risk of developing burnout and this can have serious consequences for the sustainable employability of these blue-collar workers.

## 1. Introduction

This paper focuses on the association between time perspective in relation to burnout and successful ageing of blue-collar shipyard workers in Spain with physically highly demanding work and low autonomy. This theme is important due to the ageing of the workforce, the increase in the retirement age in many western countries as well as that employers are motivated to prolong the working lives of their ageing workers.

The concept of time perspective of ageing shipyard workers in relation to burnout has so far received little research attention. Nonetheless, earlier research has pointed to the importance of examining time perspective, as it significantly influences a person’s judgments, decisions, and activities [1,2]. The working conditions within the shipyard are harsh. It is predominantly blue-collar work that is related to heavy manual and repetitive work. Shipyard workers are frequently exposed to noises of ventilation and machinery. Furthermore, the working conditions are difficult due to the high temperatures and air pollution due to the use of chemicals at the shipyard [3]. To investigate the influence of time perspective among these workers in relation to the possible development of burnout in blue-collar workers, we conducted a cross-sectional study among shipyard workers in the North of Spain. Before we address the specific research questions and hypotheses of the current study, we will pay more attention to the concept and related theory of future time perspective as well as burnout. A review found positive associations between a lower socioeconomic status (SES) indicated by income and level of education and a negative future time perspective as indicated by family income and level of education. Another study suggested that workers in higher social classes predominantly have a positive time perspective, as principal developmental tasks are more likely to be actualized at later life stages compared to those in lower social classes [4].

Zimbardo and Boyd [1] understood the term time perspective to be a propensity to concentrate on the specific time periods past, present and future, together with a favourable or unfavourable assessment of a particular length of time. They recognized five forms: *Past positive standpoint*: an inclination to concentrate on the positively appraised past; thinking about past events with pleasure, focusing mainly on the pleasant occurrences and collecting souvenirs from the past; *Past negative standpoint*: a propensity to concentrate on the negatively appraised past, to frequently remember sad incidents from the past. Principally, remembering distressing moments, traumas and dismay from the past [5].

Furthermore, *Future standpoint*: a propensity to look towards the future, formulate objectives to be met and undertakings to be done. People think about their own future with pleasure. *Present hedonistic standpoint*: a propensity to exploit enjoyment “here and now”; it is related to discounting experiences from the past; they often focus on pleasure as the major element in life, despite the chance involved. *Present-fatalistic standpoint*: a propensity to live in the present passively [6]; life depends on pre-destination, so all anyone can do is to exist in the present acquiescently.

According to Sobol-Kwapinska and Jankowski [5], it appears that a focus on the favourable past and future with an active emphasis on the present are areas which match the favourable aspects of human roles. Conversely, a fatalistic time perspective and a focus on an unfavourably viewed past are particularly negative. On the other hand, a balanced time perspective is an amalgam of a strong focus on a favourably assessed past, a medium focus on the future, a medium hedonistic focus on the present, scant fatalistic focus on the present, and scant focus on an unfavourably assessed past [1,2,5].

Sobol-Kwapinska et al. [6] mentioned that it is considered a major psychological variable related to several areas of human roles, for example well-being, healthy or risky behaviour, tendency for addictions, optimism, high self-esteem, risky driving, etc. [1,2,7,8,9].

Psychological temporality also has important effects on work experiences and emotional reactions, as well as behaviour at work [10,11,12]. In organizations, the five time perspectives pointed out by Zimbardo are important for understanding positive human behaviour, although the future perspective of time is one of the most important for explaining self-management behaviour at the workplace, like for example planning, decision making and motivation [12].

According to De Lange et al. [13], successful ageing at work is the self-management process of workers necessary to become and remain employable during their life course, indicating that employees are able to pro-actively recover and improve over time through self-management skills and actions, and with support or interventions from the work environment. Nonetheless, earlier research on successful ageing at work has pointed to the influence of changing time horizons of ageing workers and its possible impact on work motivation and outcomes.

Concerning psycho-social risk, job burnout is a condition of psychological and physical fatigue. Maslach and Jackson [14] explained it as long-lasting stress response of a person to protracted exposure to psychological and relationship stress factors in the working environment, which cover emotional exhaustion, depersonalization, and decreased individual achievement.

*Emotional exhaustion* is the central quality of burnout, it refers to feelings of extreme psychological exhaustion and absence of eagerness and vigour towards work. *Depersonalization* refers to the intentional endeavour to distance oneself from work and the manifestation of uninvolved, emotionless, and distrustful sentiments towards co-workers. *Personal accomplishment* is displayed in a diminished feeling of self-respect and furthermore an unfavourable assessment of work, incapacity to feel pleasure, fulfilment, or a sense of achievement related to performing the task [15]. It is hard work to obtain a sense of achievement when being fatigued or when one is indifferent towards the people one is helping [16].

Based on the literature, six main domains are associated with increased burnout risk: high workload, lack of control, want of reward, need for sense of community, low fairness, and different values [17]. Job burnout is a frequent issue in companies and institutions, and often occurs in high-pressure environments [18]. Furthermore, quantitative job demands such as a high workload experienced and pressurised deadlines are consistently linked to burnout, just as qualitative job demands such as role conflict and role ambiguity [16] or the absence of rewards or emotional recognition [19]. Additionally, environmental factors such as occupational noise can also lead to hypertension and burnout of blue-collar workers [20]; it is also associated with higher occupational stress [21].

According to Demerouti et al. [22], burnout is identified by high levels of psychological fatigue and negative perspectives on one’s job (disengagement). There are other factors which can influence burnout: length of experience, role, employment sector and job satisfaction [23].

Job burnout often leads to indifference to others, has a bad effect on the employee, and has an unfavourable effect on the institution [24].

Emotional exhaustion has also been found to be a significant source of physical complaints, like headaches, exhaustion, gastrointestinal issues, respiratory problems, while psychological impacts include: insomnia, symptoms of depression, use of psychotropic drugs and antidepressants, symptoms of psychological ill-health [25]. Occupational stress is often associated with emotional exhaustion [26], which is a public health issue with negative outcomes which should be avoided in the workplace [27,28,29].

Past research has found an association between a negative past time perspective and fatalistic present and negative future orientation. The role of future perspective can be a limiting (negative time perspective) but also an enabling factor (in sense of positive time perspective) [30]. Paying attention to a positive time perspective in blue-collar workers can be interesting in the field of occupational health.

In the current research, the objective is to enhance the sustainable employability of blue-collar workers at the shipyard by focusing on the mental health of these workers. In this study, we want to analyse whether the risk of developing burnout is associated with the presentation of a specific type of personality, especially a way of understanding and facing life events (in terms of perceived time perspective).

We formulated the following hypotheses:

General hypothesis: *people burned out at work are associated with a characteristic personality system based on negative past and fatalistic present*.

In addition, three specific hypotheses were created:

**Hypothesis 1:** 
*People with higher levels of emotional exhaustion will have a personality system defined by a predominance of the negative past and the fatalistic present, that is, a more negative interpretation of reality.*


**Hypothesis 2:** 
*Similarly, those subjects with higher levels of cynicism will also possess a personality system defined by a predominance of the negative past and the fatalistic present, that is, a more negative interpretation of reality.*


**Hypothesis 3:** 
*People who present higher levels of professional efficiency will possess a personality system characterized by a predominance of the hedonistic present, future, and positive past.*


## 2. Materials and Methods

### 2.1. Participants

The group of test subjects comprised 497 participants—367 men (73.8%); 130 women (26.2%)—who were workers in a shipyard in the north of Spain. The respondents were selected at random, they voluntarily agreed to participate in the study, they were given the questionnaire in person along with an informed consent form and they were asked to deliver the questionnaire within a few days. The questionnaire was only applied once. To be considered a completed questionnaire, the participant had to answer all questions on the questionnaire. This research was approved by the Ethic Committee of the Universidade da Coruna (Spain) on 22 October 2019 (69/19), organizational unit responsible for the protection of human participants.

The ages of the participants were between 20 and 69 (M = 46.62, SD = 10.79); the majority were between 38 and 55 years old. In addition to age and sex, the educational level was analysed: 15% (76 subjects) of the sample studied primary education, 50.5% (251 subjects) studied secondary education, 34.3% (170 subjects) attended university studies.

Concerning the employment situations, with respect to the types of contracts, 75.5% had a permanent contract, 11.7% a temporary contract, and 11.3% a work contract. With respect to seniority in the company, the majority were between 6 and 50 years old, with M = 19.15, Md = 17, SD = 14.26. Years of work experience, from 1 year to 50 years, M = 23.76, Md = 22 (between 14 and 35 years), SD = 12.5 (This description of the sample is summarized in Table 1).

### 2.2. Instruments

#### 2.2.1. Zimbardo Time Perspective

The Zimbardo Time Perspective Instrument (ZTPI) [1] Spanish version [31] consists of 56 items and was used to calculate five dimensions of Time Perspective: the *Past-Positive factor* indicates a nostalgic view of the past (9 items, e.g., “It gives me pleasure to think about my past”); the *Past-Negative factor* reflects an often negative, unfavourable opinion of the past (10 items, e.g., “I think about the good things that I have missed out in my life”); the *Present-Hedonistic factor* indicates an enjoyment and pleasure-seeking, risk-taking perspective on time and life (15 items, e.g., “Taking risks keeps my life from becoming boring”); the *Present-Fatalistic factor* reflects a powerless and despairing outlook on the future and life, and a view the future is predetermined and not changed by personal current actions (9 items, e.g., “Fate determines much in my life”); and the *Future factor* indicates a general inclination towards the accomplishment of future objectives and is characterized by preparation and organization (13 items, e.g., “I believe that a person’s day should be planned ahead each morning”). The internal consistency of the questionnaire used was calculated for this research using the Cronbach’s Alpha index. Regarding the ZTPI questionnaire: the negative past factor obtained a coefficient of *p* = 0.91; that of positive past *p* = 0.81; the present hedonistic of *p =* 0.878; the future of *p* = 0.83; and the fatalistic present of *p* = 0.90. The mean of the five coefficients, therefore, was *p* = 0.86. Therefore, the reliability of the questionnaire, already verified by their corresponding authors, is adequate.

#### 2.2.2. Burnout

We used the Maslach Burnout Inventory—General Survey MBI-GS [32], is a 15-item self-report measure of job burnout, which has three components: emotional exhaustion, depersonalization, and low personal accomplishment [33]. The items range from 0 (Never) to 6 (Every day). Sample items are “I have become less enthusiastic about my work”, “I have become more cynical about whether my work contributes anything” and “In my opinion, I am good at my job”. The internal consistency of the questionnaire used was calculated for this research using the Cronbach Alpha index. Regarding the MBI-GS questionnaire, the emotional exhaustion factor obtained a coefficient of 0.973; the cynicism factor of 0.833; and the professional efficacy factor of *p* = 0.86. Therefore, the mean of the three reliability coefficients was 0.887. Therefore, the reliability of the questionnaire, already verified by their corresponding authors, is adequate.

#### 2.2.3. Data Analysis

The data was coded into an Excel Program spreadsheet, and then exported to the IBM SPSS Program, version 26 (IBM, Armonk, NY, USA). Descriptive tests were requested to verify that the imported data was error-free. Next, the percentages of appearance of each category were determined in the case of the qualitative variables, and the statistics of central tendency in the case of the quantitative variables. The Alpha reliability index was also calculated using Pearson correlations both for the set of items on the Zimbardo ZTPI scale and for each of the subscales both on that scale and on the MBI-GS. Lastly, three linear regression analyses were performed, one for each of the variables of being burned out that make up each of the factors of the MBI-GS test (emotional exhaustion, cynicism and professional efficacy). The variables that make up the ZTPI personality test (negative past, present hedonistic, future, positive past, and present fatalistic) were considered as predictors. The tables of each regression analysis will allow us to determine which personality variables are present in those subjects who manifest a burnout phenomenon.

## 3. Results

Firstly, Pearson correlations between the study variables were also calculated. These correlations are listed in Table 2

As can be seen, there are high correlations between the subscales of each test, as well as between those of one test and another. Therefore, the relevant regression analyses were carried out, which are as follows.

Furthermore, linear regression statistical tests were carried out, in order to determine whether or not to verify the hypotheses. This information is reflected in Table 3, Table 4 and Table 5.

In Table 3, it can be seen that two of the variables of the analysis predict the emotional exhaustion factor: negative past and negative future. The latter predicts variability in a negative sense and coincides in its prediction with the negative past variable. Therefore, scoring highly in negative past is a predictor of emotional exhaustion. The linear regression performed presented an R coefficient of 0.396 (R^2^ = 0.157). The standard error of the estimate was 1.278, and significance occurred at 0.000.

Table 4 shows which variables from the time perspective are predictors of cynicism. In this case, the only variable that has a significant value is, as before, the negative past. Thus, the negative past predicts the two negative aspects that make up burnout. The linear regression performed showed an R coefficient of 0.332 (R^2^ = 0.110). The standard error of the estimate was 1.151, and significance occurred at 0.000.

The last of the regressions carried out (Table 5) focuses on the prediction of professional efficacy. In this case, there are two explanatory variables: again, the negative past, and positive future. Therefore, it is verified that the prediction of professional efficacy is related to positive ways of coping with time and the absence of negative views of the past.

Therefore, the results indicate how ways of considering time that emphasize the negative past predict the two negative aspects of burnout, while the positive future perspective favours the positive aspect in determining burnout: professional efficacy. The linear regression performed showed an R coefficient of 0.432 (R^2^ = 0.186; *p* < 0.01 or 0.05), values somewhat higher than those previously found. The standard error of the estimate was 0.917, and significance also occurred at 0.000.

## 4. Discussion

This study shows that shipyard workers from a large shipyard in the north of Spain have a higher risk of experiencing burnout (emotional exhaustion, personal-efficacy and cynicism), which was associated with a negative past and negative future time perspective. This higher risk of experiencing burnout can have serious consequences for the sustainable employability of blue-collar workers. The following hypotheses were accepted or rejected on the basis of these results;

**Hypothesis 1:** 
*People with higher levels of emotional exhaustion will have a personality system characterized by a predominance of the negative past and the negative future, that is, a more negative interpretation of reality.*


With regard to one of the identifying variables of being burned out at work, emotional exhaustion, the significant predictor variables were the negative past and the future, but in a negative way. The fatalistic present was not significant. Therefore, people with greater emotional exhaustion have a personality characterized by viewing a negative past and an absence of future.

**Hypothesis 2:** 
*Similarly, those subjects with higher levels of cynicism will also possess a personality system characterized by a predominance of the negative past and the fatalistic present, that is, a more negative interpretation of reality.*


The results are very similar to those obtained in the previous analysis, although slightly less significant. The only predictive personality variable was the negative past, just as it was in the previous analysis.

**Hypothesis 3:** 
*People who present higher levels of professional efficiency will possess a personality system characterized by a predominance of the hedonistic present, future, and positive past.*


The results show that professional efficacy variable is associated with a positive future and no negative past.

Firstly, the results of this study are similar to earlier research showing that a negative past orientation is associated with a negative present and future perception of the working situation. Studies have linked the past negative dimension to neuroticism [34] and to an increased negative mood [35]. In contrast, it can be assumed that past positive orientation will prevent burnout, because of its influence on the stability of the individual [2].

Secondly, a systematic review by Kooij et al. has found a positive association between a lower socioeconomic status (SES) and a negative future time perspective, as indicated by family income and level of education [4].

Thirdly, a positive association between a future time perspective and locus of control (ρ = 0.48, 95% CI 0.40 to 0.56), self-efficacy (ρ = 0.44, 95% CI 0.34 to 0.53), and self-esteem (ρ = 0.31, 95% CI 0.21 to 0.40), although the number of studies including locus of control was somewhat low (i.e., k < 10), and mainly used adolescent samples [4], and self-efficacy has been related to a higher risk of developing burnout [30]. The results showed that the Deviance from a Balanced Time Perspective (DBTP) is essential for burnout proneness, and that this influence is mediated by perceived stress and self-efficacy.

Fourth, and perhaps most importantly, our findings from earlier research showed that FTP may be modified to promote physical exercise, facilitate new skill learning, safe work behaviours, and long-term career planning [4]. Social cognitive theories propose that self-efficacy can stimulate a positive time perspective and can prevent burnout. This association has not yet been elaborated in the literature of occupational health prevention.

Admittedly, this study presents several limitations that restrict the generalization of the results. One of the limitations of this study is that we do not know if the associations with burnout are different between white collar workers and blue-collar workers, or even between countries. Earlier research in on the basis of a longitudinal study in Finland found that there were no differences between the groups. Furthermore, the job stressors were much the same in both groups [36].

Another limitation of this study is that the results are based on cross-sectional data instead of longitudinal data. Last but not least, as with all studies that involve the translation of a scale, it is possible that the equivalence problems detected are due to translation errors.

Nevertheless, this study gives insight into the possible factors that can influence exhaustion and burnout in shipyard workers. On the basis of this study, it is essential to train supervisors and workers of occupational health services at the shipyard to detect signs and symptoms of negative time perspective and lack of self-efficacy in order to prevent mental problems at the workplace. Further, it is important to analyse which working factors contribute to a negative past and negative future time perspective. Other studies have shown that a lack of autonomy and high exigency have contributed to high levels of stress and lack of job satisfaction in shipyard workers [28,29]. Earlier research has shown that blue-collar workers score lower in self-management skills than white collar workers [37]. Therefore, it is important to pay attention to the self-efficacy of blue-collar workers [13].

Furthermore, the hard environmental working conditions within the shipyard due to the constant exposure to high noises and badly ventilated areas have a negative effect on the mental and physical health of blue-collar shipyard workers [28]. Other studies point that physical work demands are a risk factor for a shortened expected working life [38] and that more men were exposed to low support at work and more physically demanding work than women [39].

These conditions should be taken into consideration in order to develop Human Resource instruments, work-disability prevention programs, and successful ageing at work practices [40] targeted at the shipyard. In addition, there is the need for organizational interventions on the basis of participatory interventions that involve employees in the process of actions taken at the workplace to improve working conditions. Together with measures to promote a healthy and safe workplace, we can conclude that more attention is needed for preventive measures to avoid mental problems due to a negative time perspective. Recent studies propose that interventions to restore the person–work fit at various levels (i.e., person, job, work group, organization, and society) should be finetuned to stimulate the self-regulation of workers [4].

## 5. Conclusions

This study shows that there is an association between experiencing burnout in the sense of (emotional exhaustion, personal efficacy and cynicism) and the variables negative past and negative future time perspective in shipyard workers from a large company in the north of Spain. These workers therefore have a higher risk of developing burnout, and this can have serious consequence for the sustainable employability of these workers. This study could also be of interest for other workers with highly physically and routine demanding jobs.

## Figures and Tables

**Table 1 ijerph-17-06905-t001:** Description of the sample.

Variable	Description
Sex	367 men (73.8%) and 130 women (26.2%)
Age	Range: 20–69. Mean: 46.62. Standard Deviation: 10.79
Education level	Primary education: 76 subjects (15%) Secondary education: 251 subjects (50.5%) University studies: 170 subjects (34.3%)
Types of contracts	Permanent contract: 75.5% Temporary contract: 11.7% Work contract: 11.3%
Seniority in the organization	Mean: 19.15. Standard Deviation: 14.26
Work experience	Mean: 23.76. Standard Deviation: 12.50

**Table 2 ijerph-17-06905-t002:** Pearson Correlations between the studied variables.

	Past Negative	Present Hedonistic	Future	Past Positive	Present Fatalistic	Emotional Exhaustion	Cynicism
Present hedonistic	0.271 **						
Negative future	−0.028	0.093					
Past positive	0.425 **	0.365 **	0.213 **				
Present fatalistic	0.480 **	0.448 **	−0.180 **	0.283 **			
Emotional exhaustion	0.374 **	0.205 **	−0.100 *	0.214 **	0.237 **		
Cynicism	0.318 **	0.178 **	−0.103 *	0.137 **	0.190 **	0.625 **	
Professional efficiency	−0.113 *	0.039	0.408 **	0.093 *	−0.048	−0.105*	−0.139 **

Note: * *p* < 0.05; ** *p* < 0.01.

**Table 3 ijerph-17-06905-t003:** Personality Predictors of emotional exhaustion.

Variables	Unstandardized Coefficients		Standardized Coefficients	t	Sig.
	B	S.E.	B		
Constant	0.180	0.757		0.238	0.812
Past negative	0.762	0.152	0.296	5.010	0.000
Present hedonistic	0.190	0.152	0.072	1.252	0.211
Negative future	−0.325	0.165	−0.102	−1.973	0.049
Past positive	0.175	0.148	0.067	1.181	0.238
Present fatalistic	0.097	0.150	0.040	0.646	0.519

Note: B: beta coefficient; S.E.: Standard Error; Sign: t: T-test; Sig.: Significativity.

**Table 4 ijerph-17-06905-t004:** Personality Predictors of cynicism.

Variables	Unstandardized Coefficients		Standardized Coefficients	t	Sig.
	B	S.E.	B		
Constant	1.265	0.676		1.871	0.062
Past negative	0.687	0.138	0.302	4.992	0.000
Present hedonistic	0.230	0.135	0.099	1.700	0.090
Negative future	−0.286	0.148	−0.102	−1.927	0.055
Past positive	0.023	0.132	0.010	0.175	0.861
Present fatalistic	−0.120	0.133	−0.058	−0.905	0.366

Note: B: beta coefficient; S.E.: Standard Error; Sign: t: T-test; Sig.: Significativity.

**Table 5 ijerph-17-06905-t005:** Personality Predictors of professional efficacy.

Variables	Unstandardized Coefficients		Standardized Coefficients	t	Sig.
	B	S.E.	B		
Constant	1.432	0.539		2.656	0.008
Past negative	−0.343	0.110	−0.180	−3.122	0.002
Present hedonistic	−0.020	0.108	−0.010	−0.182	0.856
Positive future	0.953	0.118	0.411	8.099	0.000
Past positive	0.028	0.106	0.015	0.264	0.792
Present fatalistic	0.147	0.106	0.085	1.392	0.165

Note: B: beta coefficient; S.E.: Standard Error; Sign: t: T-test; Sig.: Significativity.

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
