# Peer review of "Burnout and Time Perspective of Blue-Collar Workers at the Shipyard"

_ijerph, 2020, doi:10.3390/ijerph17186905_

Round 1
Reviewer 1 Report
The authors have done a Cross sectional study looking at the burnout and time perspective of blue-collar shipyard workers in Northern Spain This is a topic of great significance to emotional wellbeing of the workers, work performance, and their work longevity. So, I appreciate authors examining this topic. Authors have concluded that
Shipyard workers have a higher risk of burnout associated with a negative past and negative future time-perspective and the higher risk can have serious consequences for the sustainable employability of blue-collar workers.
Following are my comments and recommended revisions:
- The strength of the manuscript is that it addresses a topic of great importance to the field of occupational health. There is a dearth of studies on burnout and time –perspectives, so authors have aimed to examine this relationship.
- Authors have used the appropriate statistical methods and appropriately validates scales
- There are some grammar and diction issues that can be addressed.
- Background: please define blue and white-collar worker
- Many words are inappropriately capitalized. For example, ‘Time Perspective’.
Please correct them.
- The word Burnout is one word. Occasionally it is used as Burn-out. Please correct
- How did the authors define the completion of the survey? Answering all the questions or a certain percentage?
- How many times was the survey offered? Just once? Any reward or coercion for participating or was it strictly voluntary?
- The discussion section can be expanded. I would recommend including several paragraphs (limitations, strengths) under the conclusion to the discussion section.
- Line 239-243-The conclusions drawn are not supported by the study- authors don’t mention this under results.
Reviewer 2 Report
You have a topic that seems to be interesting. I think you have sold yourself short in presenting/explaining the theoretical foundations of the study, presenting your results, and discussing the implications.
I offer the following suggestions to strengthen the study:
- I think that you need to provide a more robust discussion of the constructs upon which you are building the study.
- You need to provide a more comprehensive presentation of your results - you need to enhance the tables.
- I would also like to see the demographic data on the sample in table form.
- Also be sure you are consistent when discussing values in the results section
- Your discussion section needs a lot of enhancement. I think that you are making very large leaps in logic about the causes of stress and burnout based without a much stronger theoretical case being built for burnout being driven by time orientation.
- You need to have someone proofread this paper for English grammar from front to back.
Reviewer 3 Report
Then I will present a series of orientations to take into account in the manuscript to improve some of its elements or, at least, try to shed light on another approach perspective. Then I will present a series of orientations to take into account in the manuscript to improve some of its elements or, at least, try to shed light on another approach perspective:
TITLE
Adequate. It is brief but gives useful information that helps to locate the subject quickly.
ABSTRACT
It would be interesting to put the number and percentage of women or men in the sample. That way other scientists can get an idea of how representative the sample is in terms of the sex/gender of the participants.
The beginning of the abstract could be modified by using an impersonal way to express the same idea. For example, "The aim of the research was to investigate the..."
In relation to the keywords, it would be better to put the keywords in alphabetical order, by following a criterion. It is recommended that one of the keywords refer to the methodology.
INTRODUCTION
It is a theoretical foundation well developed and described.
Just as three hypotheses have been put forward that are based on a general objective, it would be interesting to set out the specific objectives that relate to these hypotheses.
MATERIALS AND METHOD
Line 30. Too many brackets within others.
When describing the questionnaire, some examples of an item should be given.
The value of Cronbach's Alpha should appear.
Line 159. Regarding "Calculation", the concept "Data analysis" is more common and frequent.
RESULTS
Adequate, although it is recommended to expand the information a little more to make it clearer. There are data in the discussion section that would be specific to the results section. For example, when describing specific numerical results of the model for each hypothesis (R coefficient). Such data do not appear in the results.
It could be interesting to include Pearson`s Correlation to the analysis of the relationship between variables, previously.
The Cronbach's Alpha coefficients should be located when the instruments are described in the methodology. Therefore, since there are only two paragraphs in the results and much of the information refers to Cronbach's Alpha, it is clear that the information presented in the results is too short.
DISCUSSION
The hypotheses are very well explained, although the numerical information on the R-coefficient, etc. should be in the Results section.
The greatest adjustment that needs to be made in this section, and specifically in the manuscript, is located precisely in the discussion. For it to be a true discussion, it is necessary that each hypothesis is compared with other studies, in order to assess whether similar or opposite results appear. This aspect is carried out in the discussion of the results. Furthermore, a large part of the bibliographical references used in this section must have been previously developed in the theoretical framework, that is, in the introduction of the manuscript.
CONCLUSIONS
The conclusions themselves are appropriate.
Limitations and future lines of research could go at the end of the discussion section so that the conclusions of the study are the main focus of this section.
REFERENCES
Review the format. For example, line 265. However, it is also true that in general terms the format is good.
Having said that, it is an interesting work. It would be advisable to revise the manuscript taking into consideration the aspects mentioned. The most relevant thing that should be modified is the discussion.
Round 2
Reviewer 2 Report
Thank you for your updated work - and thank you for the clear outline of changes you made to address suggested improvements. This paper now looks much better to me. I have a few specific issues which I will outline below, but want to note that you did a great job in addressing my concerns.
Specific Issues:
Line 27 – Suggest changing wording to “…of experiencing burnout…” or "...of developing burnout..."
Line 36 – Suggest changing wording to “…ageing of the workforce, the increase in the retirement age…”
Line 149 – Suggest changing wording to “To be considered a completed questionnaire, the participant had to answer all questions on the questionnaire.” or something to that effect.
Line 155 – Suggest changing wording to “Concerning the employment situations, with respect to…”
Would like to see Table 1 formatted better.
Line 215 – Suggest changing wording to “As can be seen, there are high correlations between the..”
Lines 222-224 are a bit confusing as worded and should be reworded.
Line 257 – Suggest changing text to “… risk of experiencing burnout…” or "...of developing burnout..."
Line 280 – “Firstly”
Line 338 - Suggest changing wording to “…association between experiencing burnout in the sense…” or “…association between developing burnout in the sense…”
Also, check capitalization consistency
Line 341 – same issue as noted in line 338 above (could also be “developing burnout”)
Reviewer 3 Report
Thank you very much for your effort and time to improve the manuscript. Just one suggestion: line 277, review text editing (the final of the sentence). Thank you.
